

# EATcareFULLY
## System skanowania informacji o produktach spożywczych i analizowania historii zakupów użytkownika



**Autors**: Kacper Balbus [1]  · Marta Puz [2]  · Dawid Spałek [3]  · Mikołaj Wiora [4]  · Michał Żądełek [5]

**Supervisor:** Adrianna Kozierkiewicz

### Abstract

Przeciętny konsument nie posiada szerokiej wiedzy na temat wpływu poszczególnych produktów na jego zdrowie, więc często regularnie kupuje produkty mało odżywcze oraz takie które niosą za sobą problematyczne skutki zdrowotne. Aplikacja EATcareFULLY wychodzi naprzeciw temu problemowi oferując system do identyfikacji cech produktu poprzez skanowanie jego etykiet lub kodów kreskowych. Zaczerpnięte informacje następnie przedstawia w przyjaznej dla użytkownika formie, upraszczając i skracając czas podejmowania decyzji zakupowych. Dodatkowo pozwala ona monitorować swoje zakupy, zapewniając spis wszystkich kupowanych produktów wraz z analizą zakupów dostępną na stronie oraz w ramach raportów dietetycznych dostępnych dla każdego miesiąca. W ramach wsparcia decyzji zakupowych aplikacja oferuje system rekomendacji produktów, który poleca produkty zbliżone do kupionych, lecz o lepszej jakości. By zachęcić swoich użytkowników do zmian nawyków żywieniowych, EATcareFULLY dołącza do swojej usługi system gamifikacji, który zachęca klientów do zdrowej konkurencji w kupowaniu niemniej zdrowego jedzenia.

## 1   WSTĘP

### 1.1   Opis problemu

Powszechnym problemem wśród konsumentów jest niska świadomość dotycząca jakości spożywanego jedzenia. Ludzie niezorientowani często podejmują swoje decyzje zakupowe w oparciu o intuicje, opakowania czy też smak produktów, które widzą na półkach sklepowych. Można by przypuszczać, iż jest to efekt małej dostępności wiedzy o pożywieniu sprzedawanym w sklepach, lecz jest to błędne założenie z dwóch powodów. Przede wszystkim każdy produkt w sklepie ma opis składu oraz tabele wartości odżywczych [6]. Drugą rzeczą jest to, że przeglądarka internetowa potrafi dostarczyć konsumentowi jeszcze więcej potrzebnych informacji. Dlaczego więc ludzie tak rzadko wybierają produkty dobre dla ich zdrowia? Powodów jest kilka. Po pierwsze bardzo niewygodnym jest każdorazowe sprawdzanie opakowań produktów, gdyż są one często nieczytelne. Wiele informacji będzie wartościowe, tylko wtedy gdy poprzemy je solidną wiedzą dziedzinową z dietetyki. Kolejnym powodem może być trudność w trzymaniu się planów zakupowych, gdyż nie są one nigdzie dokumentowane. Nawet w przypadku gdy konsument zapisuje sobie liste zakupów, to nie jest w stanie w przejrzysty sposób zobaczyć czy jego dieta jest zdrowa.

### 1.2   Wizja

W kontekście biznesowym nasz produkt ma na celu rozwiązanie większości tych problemów. Głównym założeniem jest uproszczenie dostępu do informacji oraz przedstawienie je w formie najbardziej czytelnej jak to tylko możliwe. Nasz docelowy użytkownik będzie mógł w trakcie każdej wizyty w sklepie uzyskać informacje o dowolnym produkcie jaki go zainteresuje. Sam dostęp do informacji będzie możliwy poprzez skanowanie kodu kreskowego lub etykiety. Po każdym skanowaniu użytkownik dowie się wszystkiego co możliwe o produkcie w skondensowanej i czytelnej formie. Dodatkowo każdy produkt, który zdecyduje się kupić będzie zapisywany w jego osobistej historii zakupów. Aby poszerzyć wiedzę użytkownika oraz uświadomić go o regularności jego nawyków żywieniowych, nasza aplikacja pozwala na wgląd do prywatnych analiz dokonywanych na zagregowanych produktach. Takie informacje pozwolą użytkownikowi

[1] ORCID: 0009-0007-0609-4713

[2] ORCID: 0009-0001-3953-3037

[3] ORCID: 0009-0008-2890-2694

[4] ORCID: 0009-0008-4796-0830

[5] ORCID: 0009-0006-9970-1633

wyciągać sensowne wnioski dotyczące jego stylu życia. Planujemy oferować co miesięczne raporty dietetyczne dostępne w formie pdf. Gdyby użytkownik chciał, będzie mógł zabrać ze sobą raporty nawet do dietetyka, który pomoże wyciągnąć właściwe wnioski. Co więcej aplikacja będzie aktywnie wspierać użytkownika w podejmowaniu zdrowych wyborów. Oferujemy system rekomendacyjny który doradzi konsumentowi lepsze, lecz podobne produkty do tych które kupuje, oraz zapewniamy system gamifikacyjny, który ma wywołać u użytkowników zdrowe poczucie rywalizacji. Pozwoli to użytkownikowi na wgląd w tablice wyników oraz nagrodzi go stosownymi osiągnięciami, które docenią jego starania. Aplikacja będzie zawierać również opcje ustalenia preferencji żywieniowych, przez co użytkownik w swoich raportach będzie w stanie dostrzec jak daleko jest od osiągnięcia swoich celów. Preferencje będą również uwzględniane w trakcie doboru rekomendowanych produktów, co sprawi że będą bardziej dostosowane do diety użytkownika.

Zaproponowany system znacząco wpłynie na decyzje konsumenckie użytkownika. Z łatwym dostępem do informacji użytkownik będzie w stanie, w szybkim czasie sprawdzić wszystko co go interesuje, oraz dokonać korzystnego z punktu widzenia zdrowia wyboru. Liczymy, że konsument przy instalacji aplikacji zweryfikuje swoje dotychczasowe produkty pod kątem zdrowotnym a następnie, z ciekawości zacznie sprawdzać następne by odnaleźć atrakcyjne alternatywy. W tym pomoże mu również system rekomendacji, które pokaże mu, że istnieją lepsze alternatywy. Regularny użytkownik będzie stale motywowany swoimi osiągnieciami na tablicy wyników oraz swoimi osobistymi osiągnięciami. Dodatkowym czynnikiem skłaniającym użytkownika do dalszej pracy nad dietą będą szczegółowe raporty, które pozwolą mu korygować swoją dietę.

## 1.3  Prace związane z tematem

### Architektura

Ze względu na szeroki zakres funkcjonalności oferowanych przez nasz system, zdecydowaliśmy się na zastosowanie architektury hybrydowej, która łączy zalety monolitu oraz mikroserwisów. Takie podejście umożliwiło nam elastyczne korzystanie z różnych technologii, dostosowując je do specyficznych potrzeb poszczególnych komponentów.

### Backend

Główna funkcjonalność systemu, obejmująca skanowanie i wyświetlanie informacji o produktach, została zbudowana na monolicie w oparciu o framework Spring Boot w języku Java. Dzięki temu uzyskaliśmy stabilne i wydajne środowisko do przetwarzania dużej liczby zapytań użytkowników. Backend w Javie pełni również funkcję front controllera, zapewniając frontendowi jednolity punkt dostępu do wszystkich usług za pośrednictwem jednego adresu IP. Dane o produktach i użytkownikach są przechowywane w bazie danych PostgreSQL, co wynika z relacyjnej struktury danych oraz preferencji zespołu.

### Frontend

Frontend został zbudowany przy użyciu frameworka React, w architekturze Progressive Web App (PWA). Taka architektura umożliwia działanie aplikacji zarówno w wersji przeglądarkowej, jak i offline. Dodatkowe funkcjonalności, takie jak robienie zdjęć kodów kreskowych i etykiet produktów, zostały zrealizowane za pomocą bibliotek React Camera Pro i Cropper.js, co umożliwia użytkownikowi samodzielne przycinanie zdjęć jeszcze przed ich wysłaniem na serwer.

### Źródło Danych

Dane o produktach są pobierane z otwartej bazy OpenFoodFacts, która jest stale aktualizowana przez społeczność. W przypadku mikroserwisów pythonowych do komunikacji z tą bazą wykorzystano dedykowaną bibliotekę, co pozwala na optymalizację przesyłu danych i redukcję obciążenia serwera głównego.

### Mikroserwisy

Dodatkowe funkcjonalności systemu zostały zrealizowane jako osobne mikroserwisy napisane w języku Python z wykorzystaniem frameworka FastAPI. Pozwala to na modularność systemu i łatwe skalowanie jego poszczególnych elementów.

Pierwszym mikroserwisem jest system analizy historii zakupów, który generuje pliki PDF zawierające analizy danych użytkownika. W tym celu wykorzystano biblioteki Matplotlib do wizualizacji danych, Pandas do ich przetwarzania oraz FPDF do generowania dokumentów. Drugim mikroserwisem jest system rekomendacji produktów, który korzysta z SentenceTransformer do bardziej trafnego porównywania

danych tekstowych. Ostatnim mikroserwisem jest system analizy etykiet, który przetwarza zdjęcia etykiet za pomocą Tesseract OCR, a następnie analizuje tekst za pomocą modelu językowego Gemini.

## Provider Tożsamości

W systemie został zaimplementowany provider tożsamości Keycloak. Wybraliśmy to rozwiązanie ze względu na jego prosty w obsłudze i stabilny system identyfikacji użytkownika. Keycloak oferuje zaawansowane mechanizmy zarządzania tożsamością, takie jak wsparcie dla logowania jednokrotnego (SSO), obsługa protokołów takich jak OAuth2 i OpenID Connect, a także łatwość integracji z naszymi istniejącymi komponentami systemu. Dzięki Keycloak uzyskaliśmy bezpieczne i skalowalne środowisko zarządzania użytkownikami, które spełnia nasze wymagania zarówno pod względem funkcjonalności, jak i wydajności.

## Wyzwania i Rozwiązania

Podczas implementacji systemu rekomendacji głównym wyzwaniem była konieczność przetwarzania bardzo dużej bazy produktów. Rozwiązaniem okazało się ograniczenie liczby produktów w bazie oraz implementacja macierzy podobieństwa, co znacząco skróciło czas przetwarzania zapytań. W przypadku systemu analizy etykiet problemem było przygotowanie obrazu w taki sposób, aby model językowy mógł efektywnie analizować tekst. Wykorzystanie Tesseract OCR oraz odpowiednia obróbka zdjęć rozwiązały ten problem, umożliwiając precyzyjne wydobywanie informacji z przesyłanych obrazów.

# 2    WYNIKI

## 2.1    Osiągnięcia

Realizując projekt "EATcareFULLY" odnotowano osiągnięcia na różnych płaszczyznach rozwoju pisanej aplikacji. W niniejszym artykule, zdecydowano się na szczegółowy opis sukcesów wobec wymienionych niżej sekcji.

### 2.1.1    Funkcjonalności

W ramach zaimplementowanych funkcjonalności głównym sukcesem okazało się kompleksowe podejście do dostarczania informacji o produkcie użytkownikowi. Zrealizowano mechanizm zczytywania kodu kreskowego żywności za pomocą wbudowanej kamery urządzenia posiadanego przez konsumenta, jak i możliwość wpisania składającego się z cyfr kodu produktu. Ostatnia z wymienionych metod znajduje zastosowanie w sytuacji, w której skanowanie nie przebiegło pomyślnie. Takie podejście do skanowania umożliwia użytkownikowi błyskawiczne dostarczenie wszystkich niezbędnych danych skanowanego towaru, korzystając z zewnętrznej ogólnoświatowej bazy danych żywnościowych OpenFoodFacts [3].

W nielicznych przypadkach, gdy danego produktu nie ma w wymienionej wyżej bazie, zespół przygotował użytkownikowi alternatywne rozwiązanie – bezpośrednie skanowanie etykiet ze składnikami. Ta funkcjonalność po stronie korzystającego z aplikacji zakłada sfotografowanie etykiety oraz opcjonalne przycięcie otrzymanego zdjęcia w celu wyeliminowania jego nieistotnych szczegółów. Następnie taki plik zostaje przekazany do analizy, gdzie podejmowane są czynności związane ze zwiększeniem czytelności obecnych na zdjęciu informacji, aby finalnie na podstawie otrzymanego z tego procesu tekstu odpytać podłączony zewnętrzny model językowy [1]. Wspomniany model zwraca użytkownikowi niezbędne dane dotyczące potencjalnej szkodliwości produktu oraz jego klasyfikacji wobec przyjmowanych powszechnie standardów. Dodatkowo tekst etykiety jest analizowany w celu identyfikacji potencjalnie szkodliwych dodatków z numerem E (np. E122) [2]. Jeśli dany dodatek znajduje się w lokalnej bazie potencjalnie szkodliwych substancji, do wyników analizy dołączane są szczegóły dotyczące jego potencjalnego wpływu na zdrowie.

W aspekcie funkcjonalności dopełniających kompleksowość aplikacji również zanotowano satysfakcjonujące rezultaty. Za pomocą realizowanej w ramach projektu "EATcareFULLY" aplikacji istnieje możliwość zbiorczej analizy produktów z historii zakupowej dla każdego użytkownika, z opcją zarówno natychmiastowego podsumowania widocznego na ekranie aplikacji, jak i wygenerowania szczegółowych raportów dotyczących sposobu odżywiania się klienta w danym miesiącu. Co więcej, w celu wspomagania konsumenta w polepszaniu jego statystyk żywieniowych wdrożono także system odpowiadający za regularne rekomendowanie podobnych, lecz lepiej ocenionych produktów niż te dotychczas kupowane, jednocześnie dbając o podstawowe preferencje żywnościowe użytkownika. Efekt końcowy obydwu wyżej wymienionych procedur zapewnia korzystającemu z aplikacji jakościowe rekomendacje w oparciu o jego

potrzeby poprzez algorytm uwzględniający podobieństwo kategorii produktów, jak i stosownie oceniający ich walory odżywcze.

Finalnie, aby zachęcić użytkownika do regularnych, zdrowszych wyborów żywieniowych w przyszłości, zaproponowano także system osiągnięć, przypisujący użytkownikowi odznaki za osiąganie istotnych progów jakościowych oraz ilościowych, dotyczących powszechnie rozumianej oraz ustandaryzowanej zdrowej żywności. Dodatkowo każdy użytkownik omawianego systemu jest w stanie śledzić swoją pozycję w zbiorczym rankingu jakości kupowanej żywności w danym tygodniu, aby dowiadywać się, jak wypada na tle innych konsumentów oraz motywować się do lepszych wyborów w przyszłości.

### 2.1.2   Cele techniczne

W trakcie rozwoju systemu osiągnięto również istotne cele techniczne, usprawniające korzystanie z aplikacji oraz procesowanie przez nią niezbędnych danych. W tym aspekcie warto wspomnieć o zrealizowanej możliwości przetrzymywania danych tymczasowych *(ang. cache)* użytkownika, pozwalających na korzystanie z aplikacji również bez aktywnego połączenia internetowego *(ang. offline)*. Realizacja tego celu umożliwia użytkownikowi dostęp do swoich danych żywieniowych nawet w sytuacjach zakłóceń łącza z serwerem, na czym szczególnie zależało twórcom projektu. Przykład ten stanowi wzorowe wykorzystanie możliwości płynących z architektury aplikacji PWA *(ang. Progressive Web Application)* [4]. Równie znaczącym co poprzedni był cel dotyczący minimalizacji czasu uzyskania odpowiedzi na zapytania dotyczące rekomendacji oraz analizy etykiety produktu. Zważając na fakt niezwykle skomplikowanego procesu uzyskiwania oraz przetwarzania tych danych, również w oparciu o zewnętrzne narzędzia, kluczowym okazało się wdrożenie ulepszeń w zakresie przechowywania danych oraz wyboru algorytmów, tudzież wykorzystywanych technologii wspierających w newralgicznych etapach procesu. Wszystkie te działania miały na celu zapewnienie użytkownikowi płynnego operowania dostępnymi narzędziami w ramach aplikacji. Finalnie czas obydwu procesów skrócił się do zaledwie kilku sekund, co stanowi typową wartość dla innych tego typu aplikacji. Po stronie wizualnej za cel przyjęto zapewnienie spójności w wyglądzie komponentów, czcionek oraz barw, nadając systemowi wyraźny i unikalny charakter. W ramach pracy w tej gałęzi aplikacji powiodło się ujednolicenie stylu stron tak, aby stanowiły ściśle powiązaną całość. Dążenie to miało przyczynić się do wzbudzenia wrażenia w użytkowniku, jakoby miał do czynienia z jednym kompleksowym systemem zamiast zbioru różnych, niezależnych funkcjonalności. W efekcie końcowym elementy specyficzne dla systemu, takie jak logo, nazwa oraz zawartość wraz z funkcjonalnościami, zwizualizowane są spójnie i czytelnie dla użytkownika, zgodnie z jego wymaganiami do tego rodzaju aplikacji.

### 2.1.3   Silne strony

W przypadku silnych stron projektu *"EATcareFULLY"* należy wymienić gruntowne podejście do informowania użytkownika o tym co dokładnie kupuje, to znaczy jakie pozytywne oraz negatywne skutki zdrowotne niesie za sobą spożywanie danego produktu.

Koniecznym w tym temacie wydaje się przedstawienie samego procesu skanowania kodu kreskowego, który zrealizowany został wykorzystując bibliotekę html5-qrcode [7]. Wybierając opcję skanowania kodu kreskowego użytkownik zostaje przekierowany do ekranu skanowania, gdzie może wybrać, z którego aparatu chce korzystać. Po dokonaniu wyboru wystarczy, iż nakieruje aparat na kod kreskowy, obejmując go w ramkę, widoczną na obrazie 1. W momencie zeskanowania kodu przez aplikację konsument momentalnie uzyskuje widok załączony na grafice 2. Zobrazowano tam kluczowe detale dotyczące zeskanowanego produktu, takie jak ocenę w scali Nutriscore [5], oznaczenia jak i zawartość procentową składników. Pozwala to na doinformowanie użytkownika o produkcie już w trakcie zakupów, zaledwie przykładając do niego telefon z działającym aparatem.

# Scan barcode

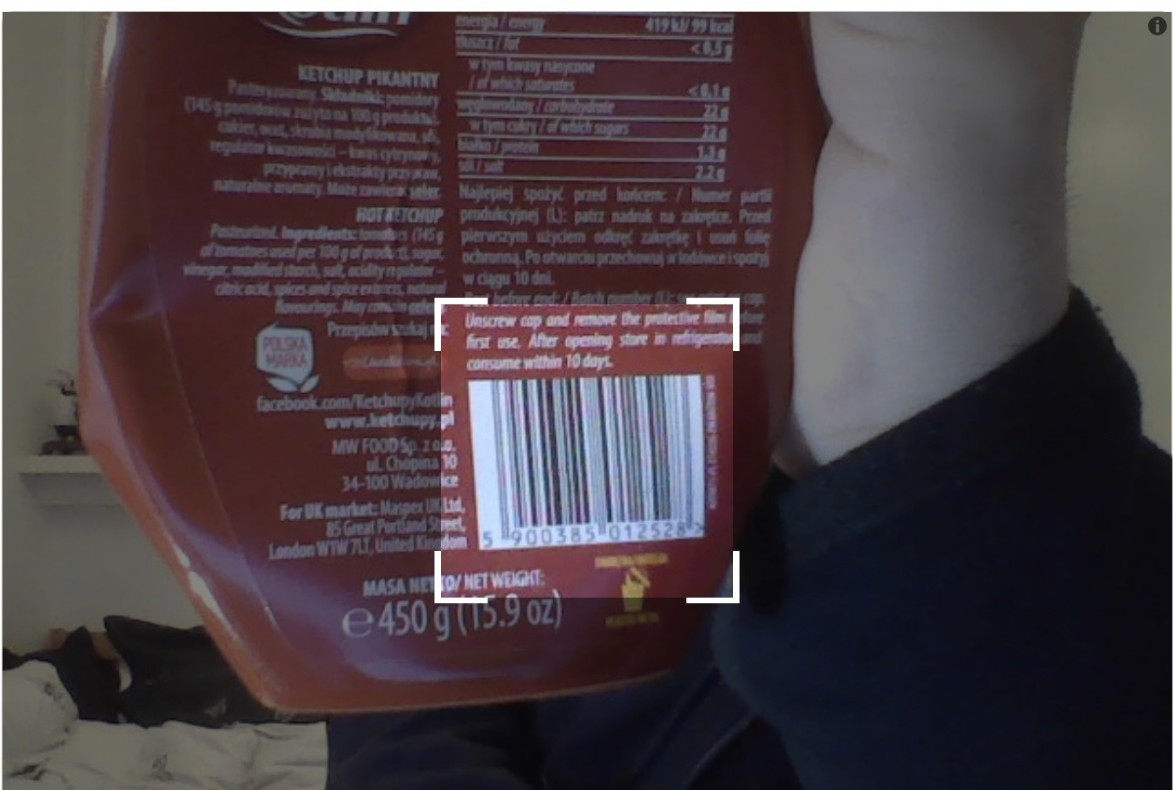

Figure 1: Ekran skanowania kodu kreskowego

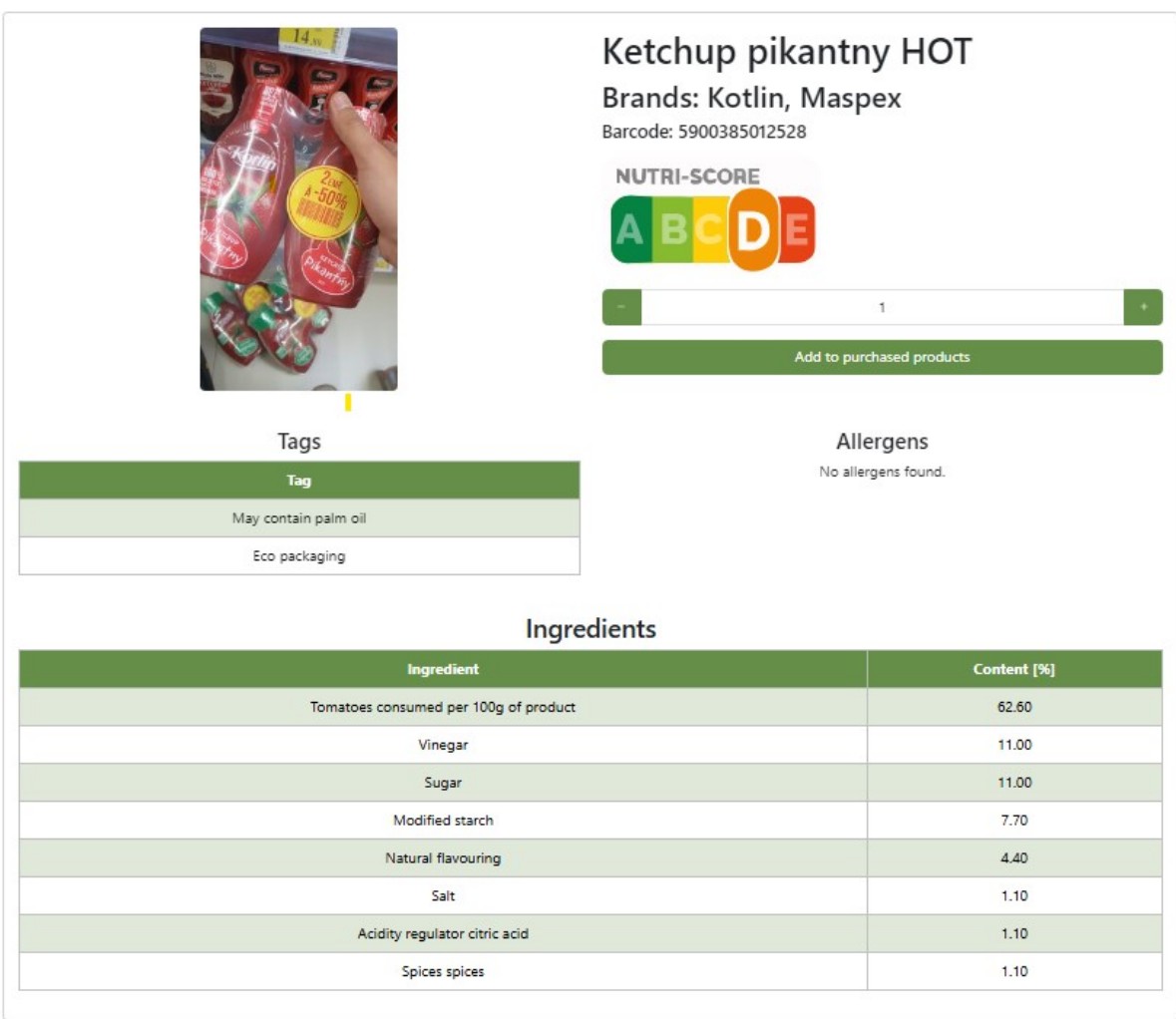

Figure 2: Szczegóły produktu po zeskanowaniu kodu kreskowego

Na załączonej niżej grafice 3 widoczny jest ekran widziany przez użytkownika tuż po wysłaniu zrobionego przezeń zdjęcia etykiety do analizy. Jak widać, mimo braku kodu kreskowego produktu w globalnej bazie, konsument nie zostaje osamotniony ze swoim problemem – wręcz przeciwnie, po chwili oczekiwania otrzymuje wszystko, co najważniejsze z załączonej fotografii, wyświetlone w przejrzystej oraz podzielonej na sekcje formie. Oznacza to, że niezależnie od miejsca zakupu, jak i od rodzaju kupowanego produktu, jedynym wymaganiem do uzyskania szczegółów towaru jest zeskanowanie etykiety z jego składnikami, którą, co warto nadmienić, posiada absolutna większość produktów na całym świecie.

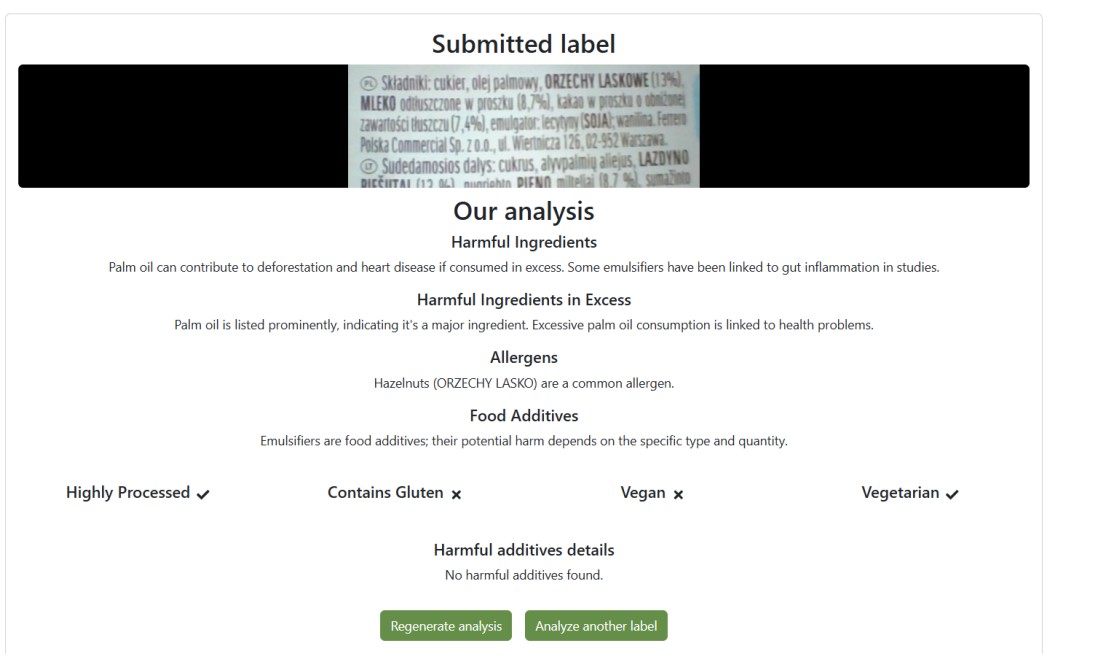

Figure 3: Wynik analizy zdjęcia przykładowej etykiety

## 2.2 Praktyczne zastosowania projektu

Projekt "EATcareFULLY" dostarcza zaawansowanych narzędzi wspierających świadome podejmowanie decyzji żywieniowych. Skierowany jest do szerokiej grupy użytkowników – od osób zainteresowanych zdrowym stylem życia po tych z określonymi potrzebami dietetycznymi. Aplikacja dostarcza korzystającemu wsparcie w trakcie, gdy jeszcze jest w sklepie – opisując produkty, które może potencjalnie kupić, i tym samym znacząco przyczynić się do jego ostatecznych wyborów. Co więcej, użytkownik może także czerpać z niej korzyści już po powrocie ze sklepu – szczegółowy panel analizy historii wspiera go w podejmowaniu przyszłych decyzji zakupowych, prowadzących do polepszenia sposobu jego odżywiania, bazując na analizie jego miesięcznej historii zakupów. Dostarczane przezeń raporty mogą posłużyć nie tylko jako materiał do własnego użytku, lecz także do przedstawienia go lekarzowi lub innemu specjaliście badającemu wpływ diety na zdrowie konsumenta. Konsument nie musi również samemu szukać lepszych jakościowo zamienników dla dotychczas nabytej żywności – aplikacja sama zaproponuje mu odpowiednie rekomendacje, w dodatku zgodne z jego deklarowanymi preferencjami. System zaproponowany przez autorów projektu wydaje się być praktycznym rozwiązaniem dla konsumentów liczących na wygodę oraz krótki czas zakupów – zachowując przy tym dbałość o detale produktów.

## 3 WNIOSKI

Dzięki zaimplementowanym funckjonalnościom, stworzony system zapewnia użytkownikowi kompleksowe wsparcie w podejmowaniu świadomych decyzji konsumenckich oraz w poprawie swojej diety. Będzie szczególnie przydatny dla osób, które nie mają specjalistycznej wiedzy na temat żywności, jak również dla tych, którym brakuje czasu na interpretacje i analizę etykiety każdego produktu podczas codziennych zakupów.

Użytkownicy mogą szybko uzyskać czytelne informacje o produktch za pomocą zeskanowanego kodu kreskowego lub zdjęcia etykiety. System automatycznie rekomenduje lepsze jakościowo alternatywy produktów oraz umożliwia analizę historii zakupów i wartości odżywczych, pomagając w długoterminowym zarządzaniu dietą. Dodatkowo motywuje do większego zaangażowania i utrzymania zdrowych nawyków żywieniowych poprzez wprowadzone elementów grywalizacji.

Za największy sukces tego projektu uważamy kompleksowe i wielopoziomowe wsparcie dla użytkownika, które zostało osiągnięte dzięki integracji wszystkich zaimplementowanych funkcjonalnośi. Każda z nich składa się na system, który nie tylko ułatwia codzienne zakupy, ale również zapewnia realną pomoc w poprawie swoich nawyków żywieniowych.

## 3.1 Kierunki rozwoju

Naturalnym kierunkiem rozwoju aplikacji może być integracja z systemami dużych sieci spożywczych, takich jak Kaufland czy Biedronka. Klienci tych dyskontów zyskaliby dostęp do przydatnych funkcji, takich jak rzetelny opis produktów oraz rekomendacja alternatyw. Dzięki temu aplikacja mogłaby dostarczać użytkownikom wartościowe informacje, bazując na rzeczywistych danych zakupowych.

Kolejnym etapem rozwoju aplikacji mogłoby być wprowadzenie nowych funkcji do użytku domowego, takich jak wirtualna lodówka. Pozwoliłaby ona użytkownikom przechowywać informacje o zakupionych produktach, łatwo nimi zarządzać oraz otrzymywać przypomnienia o zbliżających się terminach ważności. Dobrym uzupełnieniem tej funkcji byłby system przepisów, który analizowałby posiadane produkty oraz ich daty przydatności, proponując odpowiednie i optymalne dania.

## 3.2 Podziękowania

Przede wszystkim chcielibyśmy wyrazić naszą wdzięczność naszej Promotorce, dr hab. inż. Adriannie Kozierkiewicz, za wsparcie, cenne wskazówki oraz inspirację, które towarzyszyły nam podczas pracy nad projektem. Serdecznie dziękujemy również dr inż. Bogumile Hnatkowskiej za udzielone rady i pomoc w kluczowych momentach.Na koniec dziękujemy osobom realizującym projekt OpenFoodFacts za udostępnienie bogatej bazy danych, która stała się solidnym fundamentem naszego projektu.

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
