# OpenReview forum: "$EATcareFULLY$"
_pwr.edu.pl/Wrocław_University_of_Science_and_Technology/2024/ZPI_Day — Wrocław University of Science and Technology 2024 ZPI Day Submission_

### Official Review · Reviewer_of6P · 2024-12-05
**EATcareFULLY, ZPI2024 review**

**Confidence:** 4
**Significance Of Results:** 5
**Overall Quality:** 4

**Compliance With Template:**

4: High Quality – The article contains all the required sections, which are well-written and substantively correct, although minor errors or shortcomings may be present. The overall structure is clear and coherent.

**Description Of Results:**

4: High Quality – The results are described in detail and supported by usage examples or evaluations. The description is reliable but may lack full depth of analysis.

**Feedback On Consistency:**

Artykuł jest spójny, kolejność rozdziałów jest logiczna a użyty język dokładny. Należy zaznaczyć brak analizy konkurencji, co uniemożliwia rzetelną analizę innowacyjności oraz szans na wdrożenie. Jest to najistotniejszy zarzut odnośnie artykułu, a razem z niżej opisanym brakiem weryfikacji, jeden z dwóch merytorycznych.
W ramach opisu uzyskanych wyników (rozdział 2) warto eksperymentalnie zweryfikować skuteczność rozwiązania. W ramach prac inżynierskich warto pokazać mierzalne rezultaty, które określą czy aplikacja działa w sposób wystarczająco skuteczny dla danego potencjalnego klienta. W tym wypadku, naturalnym komponentem do przetestowania byłby czytnik kodów kreskowych oraz OCR. Należałoby wykonać i udokumentować testy 'w środowisku laboratoryjnym' (jako baseline, kiedy autorzy mają kontrolę nad prawidłową widocznością kodu czy tekstu) oraz w sytuacji rzeczywistej, gdy nie zawsze możliwa jest kontrola prawidłowego oświetlenia, etykieta jest uszkodzona, przesunięta lub w jakiś inny sposób odbiega od sytuacji bazowej. Ponieważ aplikacja domyślnie ma być przeznaczona dla użytkowników o różnym zaawansowaniu technicznym, warto zbadać ich poziom zadowolenia z użytkowania aplikacji.
Konieczne natomiast wydaje się uzupełnienie minimalnych wymagań odnośnie urządzenia użytkownika. Wobec braku testów skuteczności, nie sposób stwierdzić czy aplikacja jest równie skuteczna dla modeli telefonu z aparatem o niższej rozdzielczości i bez software'owej korekcji obrazu.

Opisane luki można uzupełnić wykorzystując pustą przestrzeń przy zamieszczonych obrazach (w szczególności str. 5) oraz przenosząc podziękowania do stopki.

**Potential For Development:**

Autorzy wskazują dwa potencjalne kierunki rozwoju, krótko opisując swoje pomysły. Obie propozycje wydają się przemyślane.

**Project Nature Evaluation:**

Zaprezentowana w artykule aplikacja z pewnością nosi charakterystykę inżynierską. Aplikacja wygląda na gotową do wdrożenia, jednak brak opisu jej skuteczności oraz brak analizy istniejących rozwiązań, nie pozawala na oszacowanie innowacyjności czy też skuteczności wdrożenia.
Sam artykuł posiada cechy inżynierskie i jego bardzo wysoką jakość zaburza wyłącznie brak weryfikowalności. Nie przedstawiono mierzalnych rezultatów, nie podano też linka do aplikacji ani kodów źródłowych.

**Technical Language Precision:**

4: High Quality – The language is appropriate for a technical report. Terminology is used correctly, and statements are precise, with only minor shortcomings that do not affect the overall clarity.

---

### Official Review · Reviewer_4KWy · 2024-12-06
**Ciekawa aplikacja, choć nie wiem czy posiada realne walory użyteczności.**

**Confidence:** 5
**Significance Of Results:** 3
**Overall Quality:** 4

**Compliance With Template:**

3: Average Quality – The article includes most of the required sections, but some may be incomplete, written in a general or unclear manner. The content is correct but requires further refinement.

**Description Of Results:**

4: High Quality – The results are described in detail and supported by usage examples or evaluations. The description is reliable but may lack full depth of analysis.

**Feedback On Consistency:**

Opis jest spójny, jednak brak w nim analizy porównawczej do istniejących rozwiązań z tego obszaru.

**Potential For Development:**

Aplikacja ma potencjał rozwoju, ale brak jest ewaluacji jej użyteczności.

**Project Nature Evaluation:**

Projekt ma charakter inżynierski i polega za zaprojektowaniu i implementacji aplikacji.

**Technical Language Precision:**

4: High Quality – The language is appropriate for a technical report. Terminology is used correctly, and statements are precise, with only minor shortcomings that do not affect the overall clarity.

---

### Official Review · Reviewer_sHGp · 2024-12-07
**Recenzja EATcareFULLY**

**Confidence:** 4
**Significance Of Results:** 3
**Overall Quality:** 4

**Compliance With Template:**

3: Average Quality – The article includes most of the required sections, but some may be incomplete, written in a general or unclear manner. The content is correct but requires further refinement.

**Description Of Results:**

5: Very High Quality – The results are described in detail, clearly and comprehensively, supported by thorough evaluation, analysis, and convincing usage examples. The description meets the highest substantive standards.

**Feedback On Consistency:**

Analiza problemu przedstawiona w pierwszej części artykułu, jest precyzyjnie i w pełnym zakresie rozwinięta w części opisującej rezultaty projektu. Podsumowanie nawiązuje w pełni do problemów poruszonych we Wstępie i Wynikach pracy.

**Potential For Development:**

Autorzy wskazują przyszłe funkcjonalności ale także proponują kierunki rozwoju systemy (integracja z dyskontami).

**Project Nature Evaluation:**

Autorzy poprawnie uzasadniają wybór technologii, oraz przedstawiają opis projektu/architektury w wyczerpujący sposób. Brakuje opisu procesu wytwórczego.

**Technical Language Precision:**

4: High Quality – The language is appropriate for a technical report. Terminology is used correctly, and statements are precise, with only minor shortcomings that do not affect the overall clarity.

---

### Decision · Program_Chairs · 2024-12-10

Accept (Poster)